Direct evidence of megamammal-carnivore interaction decoded from bone marks in historical fossil collections from the Pampean region

Chichkoyan Karina Vanesa 1 2
Figueirido Borja Borja.figueirido@uma.es 3
Belinchón Margarita 4
Lanata José Luis 5
Moigne Anne-Marie 6
Martínez-Navarro Bienvenido 1 2 7
1 IPHES, Institut Català de Paleoecologia Humana i Evolució Social , Tarragona , Spain
2 Area de Prehistoria, Universitat Rovira i Virgili (URV) , Tarragona , Spain
3 Departmento de Ecología y Geología, Facultad de Ciencias, Universdad de Málaga , Málaga , Spain
4 Museo de Ciencias Naturales de Valencia , Valencia , Spain
5 IIDyPCa, CONICET, UNRN , San Carlos de Bariloche , Argentina
6 Prehistory Dpt-UMR 7194 HnHp, Musée de l’Homme , Paris , France
7 ICREA , Barcelona , Spain
Knoll Fabien
Electronic publication date: 2017 May 9
Publication date: 2017
Volume: 5
Electronic Location ID: e3117
Received 2016 Nov 11; Accepted 2017 Feb 23
Copyright: ©2017 Chichkoyan et al.
Copyright year: 2017
Copyright holder: Chichkoyan et al.
License: This is an open access article distributed under the terms of the Creative Commons Attribution License, which permits unrestricted use, distribution, reproduction and adaptation in any medium and for any purpose provided that it is properly attributed. For attribution, the original author(s), title, publication source (PeerJ) and either DOI or URL of the article must be cited.
License URL: https://creativecommons.org/licenses/by/4.0/

Keywords: Quaternary, Pleistocene, Pampean region, Bone marks, Carnivore, Megamammal, Interaction

Funding: Erasmus Mundus Spanish Ministry of Economy and Competitiveness CGL2010-15326 Generalitat de Catalunya Research Group GENCAT 2014 SGR 901 This work was financed by the Erasmus Mundus grant inside the International Doctorate in Quaternary and Prehistory programme (KVCH), and by the CGL2010-15326 project of the Spanish Ministry of Economy and Competitiveness, and by the Generalitat de Catalunya Research Group GENCAT 2014 SGR 901. The funders had no role in study design, data collection and analysis, decision to publish, or preparation of the manuscript.

==============================
Pleistocene South American megafauna has traditionally attracted the interest of scientists and the popular media alike. However, ecological interactions between the species that inhabited these ecosystems, such as predator-prey relationships or interspecific competition, are poorly known. To this regard, carnivore marks imprinted on the fossil bones of megamammal remains are very useful for deciphering biological activity and, hence, potential interspecific relationships among taxa. In this article, we study historical fossil collections housed in different European and Argentinean museums that were excavated during the 19th and early 20th centuries in the Pampean region, Argentina, in order to detect carnivore marks on bones of megamammals and provide crucial information on the ecological relationships between South American taxa during the Pleistocene. Our results indicate that the long bones of megafauna from the Pampean region (e.g., the Mylodontidae and Toxodontidae families) exhibit carnivore marks. Furthermore, long bones of medium-sized species and indeterminate bones also present punctures, pits, scores and fractures. Members of the large-carnivore guild, such as ursids, canids and even felids, are recognised as the main agents that inflicted the marks. We hypothesize that the analysed carnivore marks represent the last stages of megaherbivore carcass exploitation, suggesting full consumption of these animals by the same or multiple taxa in a hunting and/or scavenging scenario. Moreover, our observations provide novel insights that help further our understanding of the palaeoecological relationships of these unique communities of megamammals.

Introduction

Reconstructing the biotic interactions between extinct organisms, including competition or predator–prey relationships, is an extremely difficult task, especially when the information available from living analogues is limited (Figueirido, Martín-Serra & Janis, 2016). This is particularly true in the case of ancient South American ecosystems, as members of the megafauna became extinct during the latest Pleistocene-early Holocene, and these groups of mammals have no living counterparts (Cione, Tonni & Soibelzon, 2009; Fariña, Vizcaíno & De Iuliis, 2013).

Megamammals from the southern portion of South America, or the Pampean (Argentinean) region, have fascinated scientists since the 18th century. Nevertheless, different studies performed to understand their palaeoecology are much more recent (e.g.,  Fariña, 1996; Bargo, 2003; Prevosti, Zurita & Carlini, 2005; Prevosti & Vizcaíno, 2006; Figueirido & Soibelzon, 2010; De Los Reyes et al., 2013; Fariña, Vizcaíno & De Iuliis, 2013; Scanferla et al., 2013; Soibelzon et al., 2014; Bocherens et al., 2016). To this respect, carnivore marks preserved on fossil bones of megaherbivores constitute an important source of information, as they represent direct evidence of predator–prey relationships, or alternatively, of scavenging activity by top predators such as strictly flesh-eating or bone-cracking hypercarnivores, respectively (e.g., Haynes, 1982; Marean & Ehrhardt, 1995; Pobiner & Blumenschine, 2003; Pickering, Egeland & Brain, 2004; Palmqvist et al., 2011; Espigares et al., 2013). Consequently, detecting the marks of biological activity preserved on the bone surfaces of Pampean megamammals, using detailed taphonomic investigations and next-generation techniques, is crucial for deciphering the ecological relationships between Pleistocene South American palaeocommunities.

Previous studies of bone surfaces performed on fossil collections housed in various museums in the Americas have revealed carnivore activity, and hence animal interaction (Haynes, 1980; Martin, 2008; Martin, 2016; De Araújo Júnior, De Oliveira Porpino & Paglarelli Bergqvist, 2011; Dominato et al., 2011; Labarca et al., 2014). Indeed, in South America, carnivore marks have been reported from different locations (Fig. 1). Specifically in the Pampean region, there is a neural apophysis of a glyptodont cf. Eosclerocalyptus lineatus (Glyptodontidae, Hoplophorini) from the Pliocene (Olavarría) with a clear carnivore tooth imprint, attributed to a giant Chapalmalania (Carnivora, Procyonidae) procyonid (De Los Reyes et al., 2013). Recently, a taphocoenosis from the margins of the Salado River, comprising remains of the equid Hippidion principale (Perissodactyla, Equidae) and some indeterminate bones with carnivore marks was associated with the dirk-toothed sabre cat Smilodon sp. (Carnivora, Felidae, Machairodontinae) (Scanferla et al., 2013). At the archaeological site Arroyo Seco 2, bones of extinct horses such as Equus sp. (Perissodactyla, Equidae) show carnivore marks (Politis et al., 2016). In Patagonia, the jaguar Panthera onca mesembrina (Carnivora, Felidae, Pantherinae) was reportedly responsible for interventions involving the ground sloth Mylodontidae (Xenarthra, Tardigrada) and Hippidion groups (Martin, 2008; Martin, 2016), and a member of Felidae produced marks on mastodont (Proboscidea, Gomphotheriidae) bones (Labarca et al., 2014) during the late Pleistocene. In Brazil, two sites have been described where the small canid Protocyon troglodytes (Carnivora, Canidae) presumably scavenged the carcasses of two mastodons, Notiomastodon platensis (Proboscidea, Gomphotheriidae), the giant ground sloths Eremotherium laurillardi (Tardigrada, Megatheriidae) and Glossotherium (Tardigrada, Mylodontidae) (De Araújo Júnior, De Oliveira Porpino & Paglarelli Bergqvist, 2011), and Haplomastodon waringi (Proboscidea, Gomphotheriidae) in the Pleistocene (Dominato et al., 2011).

Figure 1 South American map showing the three sites mentioned in the text.

In Brazil: Tanque do Jirau, Águas de Araxá. In the Pampean region, Salado River and material found in different collections of this study, Olavarria and Arroyo Seco 2. In the Patagonian region: Pilauco, Cueva Lago Sofía, Cueva del Milodón, Alero Dos Herraduras, Cueva de Los Chingues, Cueva del Puma, Cueva Fell, Alero Tres Arroyos. Image modified from Wikipedia (https://commons.wikimedia.org/wiki/File:BlankMap-Americas.svg; public domain).

In this article, we study for the first time, carnivore marks on megamammal (>1,000 kg; Cione, Tonni & Soibelzon, 2009) remains from different fossil collections recovered from the Pampean region and now housed in various institutions in Europe and Argentina. Our goal is to identify potential biological activity using taphonomic methods in order to understand predator-megaherbivore interaction within Pleistocene South American mammalian communities from the Pampean region.

Materials & Methods

In order to identify those bones showing evidence of carnivore intervention, we examined 1,976 bones belonging to the following four collections (Table 1): (i) 1,478 bones from the Rodrigo Botet collection, housed at the Museo de Ciencias Naturales de Valencia (MCNV; Spain), result of the excavations undertaken by Enrique de Carles in the northeast of the Buenos Aires province (Belinchón et al., 2009); (ii) 30 bones from the Dupotet collection, housed at the Muséum National d’ Histoire Naturelle (MNHN; Paris, France) of Pampean age from Luján City; (iii) 330 bones from the Krncsek collection, housed at the Naturhistorisches Museum Wien (NMW; Austria) that proceed from the Luján River in Mercedes City, and are identified as “Diluvium-Upper Pampean”; and (iv) 138 bones from the Ameghino collection, housed at the Museo de La Plata (MLP; Argentina), and which were extracted from a 20 m stretch along both sides of a water channel in the Canal de Conjunción (La Plata) (Ameghino, [1889] 1916 :128–129).

Table 1 Megamammal bones with museums assignation and current biostratigraphical determination.

Museum	Museum abbreviation	Taxon	Museum asignation	Biostatigraphical determination	
Museo de Ciencias Naturales de Valencia	MCNV	cf. Scelidotheriinae gen.	–	Ensenadan to Lujanian stage/age	
Muséum National d’ Histoire Naturelle	MNHN	Glossotherium robustum	Pampean	Bonarian and Lujanian stage/age	
Naturhistorisches Museum Wien	NMW	Glossotherium robustum	Diluvium- Upper Pampean	Bonarian stage/age	
Museo de La Plata	MLP	Toxodontidae	Ensenadan	Ensenadan stage/age	

These collections were gathered during various non-systematic excavations carried out in the eastern region of what is currently Buenos Aires province, in the Pampean region (Argentina), during the 19th and early 20th centuries. This is an extensive, flat geomorphological unit located in the central area of Argentina. The Quaternary was characterised by loess deposition, with different regressive and transgressive events (Fucks & Deschamps, 2008; Cione, Tonni & Soibelzon, 2009). The early and middle Pleistocene corresponds to the Ensenadan and Bonaerian Stages/Ages that were characterised by a cold and arid environment (Fucks & Deschamps, 2008; Cione, Tonni & Soibelzon, 2009). An important faunal turnover marks the boundary between the two stages, at ca. 0.5 Ma (Cione, Tonni & Soibelzon, 2009). The late Pleistocene-early Holocene corresponds to the Lujanian Stage/Age. Significant palaeoenvironmental oscillations, aeolian pulses, fluvial process and various pedogenetic events influenced this period (Tonni et al., 2003; Fucks & Deschamps, 2008; Cione, Tonni & Soibelzon, 2009). When the collections analysed in this study were originally collected, these units were included in the “Pampean Formation” (Tonni, 2011). Current biostratigraphical information (Tonni, 2009) allows the material from MCNV to be assigned to the Ensenadan to Lujanian Stage/Age and the material from MNHN and NMW to the Bonaerian and Lujanian Stages/Ages. Furthermore, in the NMW collection, the old reference to Upper Pampean is currently equivalent to the Bonarian Stage/Age (Tonni, 2011). The last record of these mammal groups comes from the Guerrero Member of the Luján Formation, deposited between 21,000 and 10,000 14C years BP (Tonni, 2009). In the case of the MLP assemblage, the presence of the notoungulate Mesotherium cristatum (Notoungulata, Mesotheriidae) among the identified species means this material can be dated as Ensenadan (Cione, Tonni & Soibelzon, 2009) (Fig. 2 and Table 1).

Figure 2 Pleistocene formations, Stage/Age and the locations of the collections over time.

Abbreviations: Ma, million years of ago. Not scaled.

To understand the natural burial conditions of the remains, we considered different types of bone surface modifications such as post-depositional fractures, the presence of original sediment or concretions, fluvial erosion, trampling, weathering, root growth, manganese spots and burning traces (e.g., Behrensmeyer, 1978; Binford, 1981; Shipman, 1981; Olsen & Shipman, 1988; Lyman, 1994; Fernández-Jalvo & Andrews, 2003; Fernández-Jalvo & Andrews, 2016). These allowed us to discard any type of intervention that could simulate carnivore activity or, if superimposed onto carnivore marks, could have indicated a previous carnivore intervention.

We follow the literature to identify whether bone marks were the result of carnivore activity (e.g., Haynes, 1980; Haynes, 1982; Haynes, 1983; Binford, 1981; Capaldo & Blumenschine, 1994; Lyman, 1994; Domínguez-Rodrigo & Piqueras, 2003; Pickering, Egeland & Brain, 2004; Domínguez-Rodrigo et al., 2012; Delaney-Rivera et al., 2009; Sala, Arsuaga & Haynes, 2014; Sala & Arsuaga, in press). As large mammal bones are too large to be ingested (Fernández-Jalvo & Andrews, 2016), we did not considered this effect as a possible agent of the marks. Furthermore, small bones tend to be splintered by the teeth of predators, making them impossible to classify either anatomically or taxonomically (Fernández-Jalvo & Andrews, 2016). Therefore, this type of fragmented material was not included in our review. The only exception was the case of the indeterminate and medium-sized bones from the MLP collection where part of the original association was conserved. Coprolites were absent in the reviewed collections.

We classified the bone marks potentially produced by carnivores into four categories (Table S1): (i) pitting and/or punctures, (ii) u-shaped elongated scratches or scores, (iii) furrowing; and (iv) spiral fractures. To investigate the body size of the potential carnivores that inflicted the marks, we used a box-plot diagram (Hammer, Harper & Ryan, 2001) to compare the size of the pitting and/or punctures from the MCNV, MNHN and MLP specimens with those published by Pickering, Egeland & Brain (2004) (various bones), De Los Reyes et al. (2013) (bone specimen Xen 30-12), and Martin (2016) (various bones); the material from NMW was excluded for the small sample size (Tables S2–S5). We follow the studies mentioned above as they allowed us to compare palaeontological and archaeological cases from the Pampean region, Patagonia, and one African case, and appreciate any similarities and/or differences with African ecosystems. Even though this information was still statistically poor, it allowed us to make some preliminary assumptions. Additionally, assigning a pit or puncture to a specific taxa is always problematic given the different factors involved (e.g., the part of the bone marked and the bite force of an animal) (Delaney-Rivera et al., 2009). Nevertheless, the overlapping of our data with the comparative cases allowed us to ascribe the marked bones to general carnivore size categories. Even though some authors have also included scores in their studies of body size (Delaney-Rivera et al., 2009; Labarca et al., 2014; De Araújo Júnior, De Oliveira Porpino & Paglarelli Bergqvist, 2011), we agree with Domínguez-Rodrigo & Piqueras (2003) that score marks relate not only with teeth size, but also the effect of the teeth being dragged over the bone surface; variability can therefore be expected from this type of marks.

We also reviewed actualistic studies describing the marks that different carnivore taxa leave when feeding and, more specifically, recent research into marks made by the members of the large carnivore guild, such as ursids (Carnivora, Ursidae), felids (Carnivora, Felidae) and canids (Carnivora, Canidae) (Table S1). Specialised bone-breaking hyenas were not considered because they were not present in South America at that time. Various studies report that ursids leave scarce to abundant teeth marks (Haynes, 1980; Haynes, 1983; Burke, 2013; Saladié et al., 2013; Arilla et al., 2014; Sala & Arsuaga, in press). In contrast, felids tend to make fewer marks on the bones since they feed exclusively on meat (Christiansen & Wroe, 2007; Sala & Arsuaga, in press), although they can leave important signs of predation (Haynes, 1983; Marean & Ehrhardt, 1995; Martin, 2008; Martin, 2016; Domínguez-Rodrigo et al., 2012; Kaufmann et al., in press; Sala & Arsuaga, in press). Finally, canids can make a great number of intervention marks (Haynes, 1982; Haynes, 1983; Yravedra, Lagos & Bárcena, 2011; Burke, 2013; Domínguez-Rodrigo et al., 2012; Sala, Arsuaga & Haynes, 2014; Sala & Arsuaga, in press). Furthermore, while felids (including Smilodon) and ursids have straighter incisive arcades, canids have curved arcades (Biknevicius, Van Valkenburgh & Walker, 1996). This shape is useful when analysing pitting and/or puncture arrangements on bone surfaces (e.g., linear or curved rows of tooth impressions).

We examined the fossil remains of the megaherbivores present in the collections with 3.5× and 12× magnifying glasses. We also used a Dino-Lite Microscope AD4113T (at magnifications of 20×–45×) and the software Dino-Lite 2.0. Both the length and breadth (major and minor axes) of the scores, pits and punctures were measured. Larger marks were measured using a caliper, and smaller ones were recorded with the measurement tool installed in the Dino-Lite. For each collection, high-resolution digital images were taken, in each museum, using a Panasonic Lumix DMC-TZ35 camera.

For the MLP assemblage we also applied the well-established archaeozoological variables MNI (Minimum Number of Individuals) and NISP (Number of Identified Specimens), as all the specimens are part of the same taphocoenosis (Lyman, 1994). While MNI was used to account for the minimum number of mammals with carnivore marks represented in the sample, the second informed the counting per taxa or skeletal part categories.

Results

We found four bones (0.2% of the total) of megaherbivores and 24 bones (1.24% of the total) of medium-sized and indeterminate species with potential carnivore intervention. In addition, a detailed description of the marks is given in Data S1. Below, we give a general overview of the most important damage found in each collection (Table 2 and Table S5) and provide general observations from the box-plot diagram (Fig. 3):

Table 2 Measurements of pits, punctures and scores.

Presence of furrowing or crenulated edges was also indicated.

Museum/specimen	Species	Element	Pitting/punctures	Scratches/scores	Crenulated	Furrowing	
MCNV (64–492)	cf. Scelidotheriinae gen.	Right tibia	4 × 3 mm/5 × 3 mm/9 × 6 mm/5 × 4 mm. Pittings on distal articular face, medial edge	(i) 20 × 10 mm. Score on distal articular face, lateral edge.
(ii) 45 × 10 × 4 mm/ 13 × 10 mm/ 20 × 13 mm. Grooves medial face of the diaphysis
(iii) 15 × 4 mm (Five marks of distal posterior face) and 15 × 5 mm (Two marks proximal posterior face).	× 	× 	
MNHN (MNHN.F. PAM 119 )	Glossotherium robustum	Left humerus	8 × 6 mm/ 7 × 7 mm/ 6 × 5 mm/3 × 3 mm. Punctures in trochlear region	(i) 45 × 10 cm groove in the condyle
(ii) 10 × 7 mm/ 15 × 6 mm/ 15 × 10 mm scores in condyle	–	× 	
MN (1908.XI.110)	Glossotherium robustum	Left distal humerus	8.5 × 6 mm	–	× 	× 	
MLP (MLP 15-I-20-32)	Toxodontidae	Femur condyle	–	Three scratches of 40 × 5 mm/ Five scratches of 15 × 5 mm	–	–	
MLP (MLP 15-I-20-36)	Indeterminate	Indeterminate	8 × 8 mm	–	–	–	
MLP (MLP 15-I-20-39)	Indeterminate	Indeterminate	4.5 × 4 mm	–	–	–	
MLP (MLP 15-I-20-40)	Indeterminate	Diaphysis	3.5 × 2 mm/ 6.5 × 4 mm	–	–	–	
MLP (MLP 15-I-20-41)	Indeterminate	Diaphysis	2 × 2 mm	–	–	–	

Figure 3 Box plot diagram showing the Log-transformed area of the pits/punctures.

Bones from MCNV 64-492, MNHNF.PAM 119, MLP, Xen 30-12 (De Los Reyes et al., 2013, Table 1), Cueva del Milodón (Martin, 2016) and Swartkrans Member 3 (Pickering, Egeland & Brain, 2004, Appendix A, column of large mammals) (Generated using the PAST program, Version 3.14; Hammer, Harper & Ryan, 2001).

Figure 4 Right tibia of cf. Scelidotheriinae gen., 64-492 from MCNV, posterior-medial view, indicating the different marks described in the text.

(A) distal epiphysis, the rectangle and zoom indicate the four linearly-positioned pits; (B) metadiaphysis with the U-shaped parallel scores circled; (C) furrowing of the distal metadiaphysis, with a circle indicating the parallel, V-shaped teeth marks on the posterior face; (D) medial face of the diaphysis with a magnified image of one of the three thick grooves; (E) furrowing of the proximal metadiaphysis.

(i) A right tibia from the MCNV (no 64-492) that corresponds to the ground sloth cf. Scelidotheriinae gen (Tardigrada, Mylodontidae). This bone presents important furrowing on both epiphyses and pits and scores on the distal epiphysis, as well as on the posterior and medial faces of the diaphysis (Fig. 4). In the box plot diagram it can be observed that the measurements of these pits slightly overlaps with the maximum sizes of large carnivores (and outliers) from Pickering, Egeland & Brain (2004) and falls within the measurements presented by De Los Reyes et al. (2013), but are slightly bigger than the Pampean case (De Los Reyes et al., 2013). Nevertheless, this discrepancy could be due the bigger pit from MCNV that seems to be enlarged by post-depositional process (Data S1 and Fig. S1). They also coincide with the smaller sizes from Cueva del Milodón (Martin, 2016);

(ii) A left humerus of Glossotherium robustum labelled MNHN.F. PAM 119 from MNHN, with pits, scores and furrowing (Fig. 5). Comparing this with the other samples reveals the same trend as for MCNV. It matches with the log area of the tibia from MCNV, but also overlaps more with the specimens in Pickering, Egeland & Brain (2004) because of the presence of smaller pits on the MNHN bone. It also coincides with the range of Xen 30-12, but has bigger and smaller log area extremes than the Pampean case (De Los Reyes et al., 2013). In addition, it compares well with the smaller marks from Cueva del Milodón (Martin, 2016);

(iii) A left distal humerus of Mylodon robustum (no. 1908.XI.110) housed at MNW with furrowing and a possible puncture (Fig. 6). The furrowed border is scalloped and part of it is flaked. This species is considered to represent Glossotherium robustum (McAfee, 2009). Although not plotted, Table S5 shows that the log area coincides with the range for the rest of the sampled material; and

(iv) At the MLP, one femur condyle from the notoungulate Toxodontidae (MLP 15-I-20-32) (Notoungulata; Toxodonta) was found with scratches (Fig. 7). Moreover, in this collection 22 long bones of medium-sized species and two further indeterminate bones have fresh fractures, scratches, punctures/pits and crenulated edges (details of these marks are shown in Table S6) (Figs. 8–10). The box plot reveals the same trend for these pits and punctures as seen in the other cases. Nevertheless, the presence of smaller marks on this sample results in greater coincidence with the Swartkrans specimens (Pickering, Egeland & Brain, 2004), and there is partial overlap with Xen 30-12 (De Los Reyes et al., 2013). However, only the outliers from MLP coincide with the smaller sizes from Cueva del Milodón (Martin, 2016), and the plot partially overlaps with those of the material from MCNV and MNHN. The smaller pits on the MLP specimens were considered together with the bigger punctures on the two indeterminate bones. Large carnivores can generate both small and large pits and/or punctures (Delaney-Rivera et al., 2009), and this may explain the variability in the marks observed here.

Discussion

The information presented above suggests that the different types of bone marks found on both megamammal and the other mammal remains were most likely inflicted by some large-sized carnivores that inhabited the Pampean region during the Pleistocene. Considering the limited evidence available from this region, the data presented here is crucial for exploring different predator–prey and/or scavenging scenarios, at a coarse scale.

Figure 5 Left humerus Glossotherium robustum, MNHN.F.PAM 119 from MNHN, anterior view, indicating the different marks described in the text.

(A) front view of distal articular face; (B) amplification of trochlear region with punctures and scratches; (C) amplification of condyle with scoring; (D) wide grooves on the lateral face.

Figure 6 Left distal humerus of Glossotherium robustum, 1908. XI.110 from MNW.

(A) anterior face; (B) posterior face, indicating the puncture; (C) amplification of the posterior rim; and (D) indication of the flaked border.

Figure 7 Condyle of distal femur of Toxodontidae, 15-I-20-32 with elongated and U-shaped scratches.

(A) lateral face; (B) anterior view with scores; (C) medial view.

The agents: pleistocene mammalian predators from the Pampean region

Several species of Quaternary carnivores have been recorded from the Pampean region. In the supplementary information, we offer a general description of these, along with some ecological characteristics (Data S2). These carnivores include ursids, felids and canids. The ursids comprise Arctotherium angustidens from the Ensenadan Stage/Age and Arctotherium vetustum, Arctotherium bonariense and Arctotherium tarijense from Bonarian and Early Lujanian times (Soibelzon et al., 2014; Figueirido & Soibelzon, 2010). In particular, the first species would have had an important capacity to feed on meat (Figueirido & Soibelzon, 2010). Felids are represented by three hypercarnivorous species: Smilodon populator, Puma concolor and Panthera onca (Christiansen & Harris, 2006; Prevosti & Vizcaíno, 2006; Bocherens et al., 2016). While the first two had some bone marking capacity, the third would have been capable of inflicting more damage (Van Valkenburgh & Hertel, 1993; Marean & Ehrhardt, 1995; Antón et al., 2004; Martin, 2008; Martin, 2016; Muñoz et al., 2008; Binder & Van Valkenburgh, 2010; Domínguez-Rodrigo et al., 2015; Kaufmann et al., in press). Finally, several pack-hunting and/or scavenging canids were present at the time, including Theriodictis platensis (and its sister taxon “C”. gezi) in the Ensenadan (Prevosti & Palmqvist, 2001; Prevosti, Tonni & Bidegain, 2009), various Protocyon species throughout the Pleistocene (Prevosti, Zurita & Carlini, 2005; Prevosti & Schubert, 2013; Bocherens et al., 2016), Canis nehringui (currently recognised as a junior synonym of C. dirus, (Prevosti, Tonni & Bidegain, 2009), and Dusicyon avus in the late Pleistocene (Prevosti & Vizcaíno, 2006).

Figure 8 Bone shafts showing carnivore intervention from MLP.

(A) MLP 15-I-20-35 with spiral fracture, amplifications of the internal notch and the cortical face with scoring; (B) MLP 15-I-20-34 with spiral fracture, notches can be observed on the medullar face, amplification of light pitting in the cortical face; (C) MLP 15-I-20-33 with spiral fracture

Figure 9 Bone shafts showing carnivore intervention from MLP with spiral fracture and magnification of crenulated edge.

(A) MLP 15-I-20-37; (B) MLP 15-I-20-38.

Figure 10 Indeterminate fragment of bone with puncture and amplification of the puncture with Dino-Lite measurements.

(A) MLP 15-I-20-36; (B) MLP 15-I-20-39.

It is clear that carnivores with an important capacity for bone modification and/or consumption would have been responsible for the various marks observed. Even though felids such as Smilodon or Puma could have produced some bone-damage, as observed in some studies (Van Valkenburgh & Hertel, 1993; Marean & Ehrhardt, 1995; Muñoz et al., 2008; Kaufmann et al., in press), their reduced bone-breaking potential rules them out as the principal generator of the feeding traces recorded. Furthermore, it is worth mentioning that the highly specialised viscera-eating dentition of the dirk-toothed Smilodon would have prevented this animal from feeding on carrion unlike other scimitar-toothed predators (e.g., Homotherium) (Palmqvist et al., 2007).

Identifying potential agents of the megamammal tooth-marks

Based on the box plot comparisons (Fig. 3), the marks on the samples in this study best match those made by the giant Pampean Chapalmalania (De Los Reyes et al., 2013). This procyon had previously been compared with a bear, although according to De Los Reyes et al. (2013) the cranial configuration is more similar to that of hyenas. From the information presented by Pickering, Egeland & Brain (2004), it seems that the damage inflicted also coincides to some degree with that made by large African carnivores, such as large canids, spotted hyenas and lions, or the smaller marks realised by Panthera onca mesembrina (Martin, 2016). These African species correspond to sizes 2 or 3 in the Bunn ranking (1986). Cross-referencing these sizes with the Pampean carnivores, they coincide with several ursids, felids and canids, although some Pampean species were larger, such as Smilodon populator, size 4, and Arctotherium angustidens, size 5 (Table 3). Moreover, the reports from the various South American sites involving pitting and/or punctures show a similar range of values as seen in this study (Table 4). Most of this information could not be plotted, as the number of marks found at each site was too low to be able to input them into the calculation. Nevertheless, it can be observed that the majority range from 5 to 10 mm in size (those from Cueva del Milodón are larger, as shown in the box-plot). According to this data, different members of the Pampean large-carnivore guild would have produced the bone damage observed on the samples from the various museums. To determine which carnivores were involved, we must relate the marks to the types of bone damage generated by the potential ursid, felid and canid taxa.

Table 3 Body size categories for Pampean carnivores (based on Bunn, 1986).

Pleistocene Pampean carnivores	Body size (in kg)	Body size categories	
Dusicyon avus	14	Size 1	
Protocyon	20–25	Size 2	
Canis nehringui	32	Size 2	
Theriodictis platensis	37	Size 2	
Puma concolor	47–50	Size 2	
A. vetustum/ A. bonariense/ A. tarijense	110 a 140	Size 3a	
Panthera onca	120	Size 3a	
Smilodon populator	220–360 up to 400	Size 3b/ Size 4	
Arctotherium angustidens	>1,000	Size 5	

Table 4 South American sites with reported dimensions of pitting and/or punctures (as cited in the original publication).

Site	Species	Carnivore	Puncture/pitting size (in mm)	References	
Olavarría	cf. Eosclerocalyptus lineatus (Hoplophorini)	Chapalmalania	ML 8.67/MW 4.38/Area mm2 33.93*	De Los Reyes et al. (2013)	
			ML 11.07/ MW 4.32/Area mm2 45.56*	De Los Reyes et al. (2013)	
			ML 7.98/ MW 1.95/Area mm2 10.92*	De Los Reyes et al. (2013)	
			ML 6.97/ MW 4.63/ Area mm2 30.98*	De Los Reyes et al. (2013)	
			ML 8.83/ MW 1.93/Area mm2 13.02*	De Los Reyes et al. (2013)	
			ML 7.82/ MW 2.80/Area mm2 17.12*	De Los Reyes et al. (2013)	
Arroyo Seco	Equidae	–	Average: MA A (long) 7.383/ MI A (wide) 5.727	Politis et al. (2016)	
Cueva del Milodón	Mylodon darwini	Panthera onca mesembrina	12.27 diameter	Martin (2008), Martin (2016)	
			4.34–9.05	Martin (2008)	
			41.63 × 30.36*	Martin (2016)	
			23.37 × 21.86*	Martin (2016)	
			7.10 × 5.01*	Martin (2016)	
			55.30 × 40.29*	Martin (2016)	
			10.61 × 7.46*	Martin (2016)	
			6.13 × 5.14*	Martin (2016)	
			15.09 × 4.40*	Martin (2016)	
			17.56 × 13.43*	Martin (2016)	
			7.99 × 8.64*	Martin (2016)	
			5.17 × 4.99*	Martin (2016)	
			6.84 × 8.30*	Martin (2016)	
Cueva de Los Chingues	Hippidion saldiasi	Panthera onca mesembrina	9 × 7.60	Martin (2008)	
			8.13 × 4.79	Martin (2008)	
			4.9 × 4.2	Martin (2008)	
Pilauco	Gomphotheriidae	Felidae	10.24 × 11.71	Labarca et al. (2014)	
			8.84 × 9.71	Labarca et al. (2014)	
Águas de Araxá	Haplomastodon waringi	Protocyon troglodytes	Average diameter 5	Dominato et al. (2011)	
			Average diameter 6	Dominato et al. (2011)	
	
Notes.

ML Maximum length

MW Maximum width

MA A Major axis

MI A Minor axis

Measurements marked with * were used for comparative purposes.

The MCNV cf. Scelidotheriinae gen. tibia is the bone that presents the most significant carnivore interventions. A combination of pitting, scratches and important furrowing was observed, on both the epiphyses and medial faces. Even though all three groups of carnivores were capable of leaving these types of marks, certain characteristics allow us to relate this damage to ursids. In particular, the group of aligned pits imprinted on the medial rim (Fig. 4A) of the distal epiphysis is planar that could indeed have been made by the premolars or molars of ursids (Haynes, 1983). In contrast, the parallel, V-shaped tooth marks on the posterior face (Figs. 4C and 4E) could be related to a series of incisors and canines and would coincide with the dragging action of a straight incisor arcade (Biknevicius, Van Valkenburgh & Walker, 1996). On the other side, the parallel scores, like those seen on the distal metadiaphysis (Fig. 4B), are also generally characteristic of ursids (Haynes, 1983; Saladié et al., 2013). In addition, the intensive furrowing coincides with the bone-breaking capacity of this animal (Soibelzon et al., 2014). Other damage typical of ursids observed on the tibia includes the elongated gouge on the lateral side of the articular face (Fig. 4A) and the quadrangular-shaped grooves on the medial face of the diaphysis (Fig. 4D) (Burke, 2013; Saladié et al., 2013). That being said, these grooves, and the gouges observed on the distal metadiaphysis, do not have the regular walls and bottoms characteristic of ursid marks (Saladié et al., 2013). Also, according to current research, this damage should be superficial, a feature not observed on this bone (Haynes, 1983; Saladié et al., 2013). To this respect, some authors suggest that the damage produced by ursids is less intense than that inflicted by other groups (Haynes, 1983; Arilla et al., 2014; Sala & Arsuaga, in press), a pattern not observed here. Consequently, more than one animal may have participated in imprinting the complex and producing the marks observed on this tibia. If that is the case, Panthera onca could have been involved, too. This species also possessed straight incisive arcades (Biknevicius, Van Valkenburgh & Walker, 1996) that could have produced the elongated V-shape marks (Haynes, 1983) on the posterior face. The important furrowing noticed at both ends of the bone is also consistent with this felid’s damage-producing capacity (Martin, 2008; Martin, 2016; Domínguez-Rodrigo et al., 2015).

The humerus of Glossotherium robustum housed in the MNHN has suffered less bone loss than the tibia. Feeding marks on this element have several characteristics that could indicate its consumption by Arctotherium. As observed on the tibia, the short, wide scratches present on the condyle and the wide, elongated, superficial pitting, agree with actualistic studies of ursid marks (Figs. 5A–5C) (Haynes, 1983; Burke, 2013; Saladié et al., 2013). Nevertheless, the presence of V-shape punctures in the trochlea (Fig. 5B), characteristic of felids rather than ursids, means that other taxa, such as Panthera onca, cannot be ruled out (Haynes, 1983). Both groups were capable of furrowing the epiphysis (Martin, 2008; Arilla et al., 2014; Domínguez-Rodrigo et al., 2015) as observed on the trocheal part of the bone (Fig. 5D).

The furrowing on the MNW Glossotherium robustum humerus is more ambiguous than the marks on the other two bones, since various taxa could have inflicted this type of damage on cancellous bone (Figs. 6A–6D). The cusp that made the puncture could have been on a secodont tooth from a felid or canid (Fig. 6B). Both these groups have the capacity to damage and destroy cancellous tissue, although canids leave fewer marks on mammals larger than 400 kg (Yravedra, Lagos & Bárcena, 2011). Patagonian sites with important furrowing in Mylodontidae bones, attributed to Panthera onca mesembrina, could provide an important parallel (Martin, 2008; Martin, 2016) when considering the types of marks that jaguars can make on limb bones, as seen in this case.

The marked femur of Toxodontidae from the MLP must be integrated with the other evidence from the taphocoenosis in order to interpret which carnivore species was involved. Of the 138 bones studied from this site, 61.59% (NISP: 85) belong to indeterminate species, while the remaining 38.40% (NISP: 53) were identified to genus level. Among these, equids are the most common, accounting for 36.53% (NISP: 19) of the identified elements. Megamammal bones are the second most widely represented group, with 30.76% (NISP: 16). The assemblage predominantly comprises appendicular skeletal elements (73.92% or NISP: 102). Axial and planar bones contribute only 13.77% (NISP: 19) and indeterminate fragments account for 12.31% (NISP: 17). Of the carnivore-marked bones, 88% (NISP: 22) are indeterminate diaphysis of the long bones mentioned above Table S6), coinciding with the general abundance of limb elements. Carnivore-marked bones represent only 18.11% (NISP: 25) of the total assemblage. The low proportion found at this site could have been influenced by its location in running water. As explained by Ameghino, ([1889] 1916) the material from this site was scattered along a 20 m stretch on both sides of a channel. Therefore, the current may not only have dispersed the primary association, but also mixed it with bony remains not originally consumed by the carnivore/s involved. This may also have influenced the skeletal assemblage, including the paucity of axial parts, resulting from density-mediated destruction or the winnowing of lighter axial bones. Nevertheless, the fact that carnivores mark 18.11% of the bones also indicates that a basic level of primary association remained when this material was collected. The presence of the Toxodontidae femur and other medium-sized bones with carnivore marks indicates that a MNI of 2 animals were consumed in the location itself. In addition, the dominance of fractured long bones could, partly, have been the result of carnivore activities that transported limbs to this area. Consequently, the carnivore/s involved in the formation of the collected assemblage must have had the capacity to break long bones and/or the ability to predate upon megamammals. In this sense, given the absence of specialised bone-crushers in the Americas, some type of canid may have been responsible for the described interventions. It is likely that either Theriodictis platensis or Protocyon scagliorum from the Ensenadan Stage/Age generated these marks, as also inferred for the Brazilian cases (De Araújo Júnior, De Oliveira Porpino & Paglarelli Bergqvist, 2011; Dominato et al., 2011).

In any event, although the proportion of carnivore marks that we have found on bones of megamammals is relatively low, this precludes the conclusion that the sites where the remains were originally collected represented the den of a hypercarnivore or bone-cracking species.

Other potential carnivores specialising in medium-sized and/or small taxa, such as Canis nehringui or Dusicyon avus, could have fed on the megaherbivore community during the late Pleistocene (Prevosti & Vizcaíno, 2006; Prevosti, Tonni & Bidegain, 2009). At ca. 14.000 cal yrs BP (Politis et al., 2016) Homo sapiens also became part of the carnivore guild. Humans not only scavenged megamammal carcasses (Politis et al., 2016), but were also more successful hunters of these animals than the existing carnivores (Cione, Tonni & Soibelzon, 2009).

Megamammal carcass consumption during the Pleistocene

Considering the skeletal elements, bone mark locations, and the level of use of the bones, it seems most likely that these marks represent the final stages of megamammal carcass consumption.

(i) Marks on the tibia and the humeri are situated on the epiphysis, both the articular surface and metadiaphyses. In a hunting event, carnivores that have access to a large mammal usually begin to feed on the abdominal part, later moving to femoral muscle masses, leaving some marks on the distal epiphyses and diaphyses (Haynes & Klimowicz, 2015). Forelimbs are usually consumed later, since the skin is harder in these areas (Haynes, 1982; Haynes & Klimowicz, 2015). The same usually happens with lower limb bones, such as the tibia, due to their smaller quantities of meat (Haynes, 1982; Blumenschine, 1986; Haynes & Klimowicz, 2015). The intense gnawing of the cf. Scelidotheriinae gen. tibia, both on the distal epiphysis and medial face of the diaphysis, as well as, to a lesser degree, on the proximal epiphysis, implies that this element was fully exploited. The presence of marks on the diaphysis indicates that even the hardest part of the shaft was utilised. The same is true for both Glossotherium robustum humeri. The damage to the distal epiphyses was inflicted in subsequent stages and not at the beginning of the consumption sequence. The presence of furrowing on the three elements implies that the various carnivores involved were consuming a substantial amount of bone. In the case of the MLP assemblage, the dominance of broken long bone diaphyses indicates access to within-bone nutrients, relating to the last stages in the consumption sequence (Binford, 1981; Haynes, 1982; Blumenschine, 1987; Capaldo & Blumenschine, 1994).

(ii) Intensity of carcass use is related to resource availability (Haynes, 1980; Haynes, 1982; Van Valkenburgh & Hertel, 1993; Delaney-Rivera et al., 2009), the size of the hunting pack (Van Valkenburgh et al., 2016), or multiple carnivore taxa involvement (Pobiner & Blumenschine, 2003; Delaney-Rivera et al., 2009). In general terms, large animal tissue is usually conserved for longer once dead (Blumenschine, 1987) and their bones have fewer marks than seen on bones of smaller species (Yravedra, Lagos & Bárcena, 2011; Domínguez-Rodrigo et al., 2015). As the easy-to-access meat is consumed, carnivores tend to eat the remaining parts of the carcass and inflict more significant damage to the bones (Binford, 1981; Haynes, 1982; Blumenschine, 1986; Pobiner & Blumenschine, 2003; White & Diedrich, 2012; Haynes & Klimowicz, 2015; Sala & Arsuaga, in press). Thus, marks on articulation surfaces could indicate that the bone held only a small amount of meat when the intervention took place. This is the case of the cf. Scelidotheriinae gen. tibia from the MCNV, the Glossotherium robustum left humerus from the MNHN, and the Toxodontidae femur from the MLP (along with other broken bones). The same hypothesis can be proposed for the Glossotherium robustum humerus from the MNW, although in this case, a lack of marks on the articulation surface could indicate that the bone was still attached to the rest of the limb. In general, the intensity of the marks and fractures observed indicates advanced stages of modification (Haynes, 1982; Sala & Arsuaga, in press).

The described feeding traces therefore appear to indicate that during the Pleistocene, different species within the large carnivore guild would have accessed and consumed megamammal bones and/or the marrow of medium-sized animals, in the final stages of a consumption sequence. Although discussion of how the animals were predated is difficult without more contextual information, given the multiple possibilities for carnivore exploitation of megamammal carcases (Pobiner & Blumenschine, 2003), two possible extreme scenarios are considered here: the marks described resulted from a first access (hunting) event and/or secondary access (scavenging) activity. The first case would involve the same group of carnivores killing and consuming the edible muscle tissues and then exploiting bones and within-bone nutrients. Early access to the carcass of an animal that had died a natural death by the same carnivore group can be also included in this situation (Blumenschine, 1986). Alternatively, after the death of the animal (either from natural causes or hunting activities), various carnivore taxa could have fed on a single carcass. In this second situation, one group would have consumed the primary edible tissues of the bony elements, and, at a later stage, the bones and marrow would have been exploited by other carnivores.

These interventions resulting from hunting and/or scavenging events indicate that in both cases, megamammal carcasses were completely exploited by various members of the large-sized carnivore guild in the region. Our samples belong to different time periods within the Pleistocene (Fig. 2 and Table 1). This provides weak but positive evidence suggesting that consumption of edible tissues as well as the bony elements and/or marrow by different carnivore groups was a pattern that occurred repeatedly throughout that period. Full exploitation of carcasses is expected, at least periodically when food is scarce and/or more carnivore species are present, as has been proposed for other American ecosystems such as Rancho La Brea (Van Valkenburgh & Hertel, 1993; Binder & Van Valkenburgh, 2010; Van Valkenburgh et al., 2016). Thus, it seems likely that temporal palaeoenvironmental stressors would have influenced the richness of Pampean megamammal communities (Cione, Tonni & Soibelzon, 2009), acting as cyclic, top-down pressures stimulating interspecific and intraspecific competition for the carcasses, resulting in the complete consumption of them.

Conclusions

Four megaherbivore fossil bones, 22 bones of medium-sized species, and two indeterminate bones with carnivore marks were studied from European and Argentinean collections of Pleistocene remains from the Pampean region, collected during the 19th and early 20th centuries. The marks were predominately identified on appendicular bones. After internal organs and muscles are consumed, limb bones are the richest parts with regard to within-bone nutrients, and in particular, the epiphyses are the easiest to penetrate by gnawing (Binford, 1981; Dominato et al., 2011; Labarca et al., 2014). Analysis of the punctures and pitting shows that these partially overlap with the range of bigger marks made by large carnivores from African environments, the smaller markings of Panthera onca mesembrina, and they are comparable with the giant Chapalmalania from the Pliocene of the Pampean region (Pickering, Egeland & Brain, 2004; De Los Reyes et al., 2013; Martin, 2016). Moreover, our measurements generally agree with the information reported from other South American sites (Martin, 2008; Dominato et al., 2011; Labarca et al., 2014; Politis et al., 2016). Consequently, it is likely that different members of the Pampean large-carnivore guild produced the marks described in this study. We interpret the data presented here as indicating the fact that ursids, canids, and possibly felids would have consumed the soft and hard tissues, inflicting various tooth marks, including pits, punctures, and scratches, furrowing bone epiphyses, and even breaking the diaphyses of long bones in order to access the marrow. These latter represent the final stages of carcass exploitation, given that the marks described on the epiphyses and diaphyses were not inflicted when bone still held large quantities of meat.

Considering that there is little information on carnivore marks from the region, as this type of evidence is still scarce, the few remains presented here significantly increase our knowledge of palaeoecological relationships in the Pampean region. The marked bones indicate that the megamammal carcases were fully exploited. This type of evidence has been recorded in the Pliocene (De Los Reyes et al., 2013) and, according to the evidence presented here, continued periodically throughout the Pleistocene. Consequently, temporal shifts in prey availability would have influenced predator–prey and/or scavenging dynamics, increasing competition for carcasses and resulting in the consumption of bone and within-bone nutrients by the same or multiple taxa. Pleistocene large mammal communities would have developed different trophic levels with multiple competitive species, allowing them to persist through time and overcome different palaeoclimatic fluctuations. This situation lasted until the late Pleistocene-early Holocene when many megafaunal extinctions occurred (Van Valkenburgh et al., 2016).

Current taphonomic methods allow new results to be obtained from historical collections. In this study, different types of carnivore marks inflicted on megamammal and other mammal bones were measured and categorised. Interpreting these with the help of current ecological information sheds light onto the palaeoecological relationships of native Pampean mammal communities from the Pleistocene. This novel perspective offers new insights into the development of future systematic fieldwork. Both collection and field-based research will provide crucial information on the evolution of the Pleistocene ecosystems of the South American Southern Cone.

Supplemental Information

Figure S1 Pit of 9 × 6 mm located on the articular face of MCNV 64-492

(A) Medial border where manganese spot abruptly ends. (B) Lateral border where the pit edge protrudes inwards.

Click here for additional data file.

Table S1 General characteristics of considered carnivore marks and their relationship with each carnivore group

Click here for additional data file.

Table S2 Calculations using the information from De Los Reyes et al. (2013)

Click here for additional data file.

Table S3 Calculations using the information from Martin (2016)

Click here for additional data file.

Table S4 Calculations using the information from Pickering, Egeland & Brain (2004)

Click here for additional data file.

Table S5 Calculations of area and log area for MCNV, MNHN, MNW and ML

Click here for additional data file.

Table S6 Carnivore marks registered in MLP

Click here for additional data file.

Supplemental Information 1 Supplementary text

Click here for additional data file.

Data S1 Description of carnivore marks

Click here for additional data file.

Data S2 Description of Pampean carnivores

Click here for additional data file.

We thank Christine Argot and Guillaume Billet (MNHN), Ursula Göhlich (NMW), and Marcelo Reguero and Martín De Los Reyes (MLP) for allowing access to the collections under their care. KVCH thanks Noel Amano from IPHES who helped with the PAST program and development of the graphics. We are especially grateful to Sam Arman, Briana Pobiner and an anonymous reviewer, whose comments, suggestions and critiques have improved this manuscript and given new insights into several ideas.

Additional Information and Declarations

Competing Interests

Author Contributions

Data Availability

The authors declare there are no competing interests.

Karina Vanesa Chichkoyan conceived and designed the experiments, performed the experiments, analyzed the data, wrote the paper, prepared figures and/or tables, reviewed drafts of the paper.

Borja Figueirido wrote the paper, reviewed drafts of the paper.

Margarita Belinchón and Anne Marie Moigne analyzed the data.

José Luis Lanata conceived and designed the experiments, analyzed the data.

Bienvenido Martínez-Navarro conceived and designed the experiments, analyzed the data, wrote the paper, prepared figures and/or tables, reviewed drafts of the paper.

The following information was supplied regarding data availability:

The raw data is included in the figures and tables in the manuscript and also in the Supplemental Information.

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
