# Peer review of "Direct evidence of megamammal-carnivore interaction decoded from bone marks in historical fossil collections from the Pampean region"

_PeerJ, doi:10.7717/peerj.3117_

## Round 0.1 · original submission · Major Revisions

Your manuscript has now been seen by three referees. You will see that, while they find your work of interest, they have raised points that need to be addressed by a major revision. All three referees have provided additional files for you.

I therefore, invite you to revise your manuscript, taking into account these points. With the revised version of your manuscript, I invite you to provide a "Response to referees" document in which you list all suggestions and corrections, together with your response to each.

As the English language in your text would benefit from improvement for clarity and readability, I strongly suggest that you either ask a colleague whose native language is English to proofread your manuscript or that you use an English language editing service.

I will send the new version of your manuscript out again for review to the three reviewers.

·

Basic reporting

There are inconsistencies in the tense used throughout. The language is also often poor, ambiguous and with numerous spelling errors. In general the paper feels 'loose', and needs further refining. There are also large sections of text, such as the descriptions of marks found which could be summarised, and spelt out in full detail in supplementary information. Markings on the numerous smaller bones should also be documented in these SI.

Experimental design

The experimental design appears well established. It may be improved though by indicating how the markings were distinguished from taphonomic markings from excavation, particularly given the age of the collections.

Validity of the findings

There are several leaps of faith in interpretation that I feel need to be addressed. These mainly are in reference to the ecosystem-level interpretations of behaviour from only a small proportion of marked bones.

Additional comments

Overall good quality and important work.

Reviewer 2 ·

Basic reporting

The text is written in a clear English language. Literature is rich enough, well adapted to the topics and well referenced. When I found a lack of references, I noticed it directly on the draft. The structure of the manuscript is conformed to the PeerJ standards. Figures are of good quality. We only suggest to the authors to orient bones in a similar way for all the photo plates (Proximal epiphysis on the left and distal epiphysis on the right) and noticed that Figure 7 is blurred and needs to be renewed.

Experimental design

Research question is clear enough. The main point of this paper is to provide an overview of all the data on the megammal-carnivore interactions in the Pampean region available in the various and dispersed historical collections in Europe and South America. Considering this question of south-american carnivore ecological niches, this article fits well with the biological scope of the journal. The methods used for this taphonomical analysis are well adapted to the topic. In the Materials & Methods section, authors used recent references and fully detailed the context of their study (analytical equipment, corpus, etc.). Nevertheless, I would suggest to the authors to add a paragraph dedicated to their method of taking tooth-marks dimensions. Those latter may be in mm and references are lacking (for example Pickering et al., JHE, 2004). Moreover, this study also lacks statistical analyses of those dimensions. As the knowledge about the Pleistocene carnivores impact on megammals in the Pampas is still poor, that information about the identification of carnivore-activity would be of interest.

Validity of the findings

Data given here are well described and analyzed and thus robust enough for any replicative analysis. Moreover, figures and tables ensure data reliability to all specialists in the field. Conclusions are rather cautious and take into account the limits of the very small corpus.

Additional comments

This paper is an interesting contribution to the knowledge of the predator-prey interaction during Pleistocene times in the pampas region (South America). Despite the very few remains available for that study (NR=4) yielded by various historical collections, authors succeeded in providing the maximum amount of information. At my sense, the main interest of this study, in addition to give an exhaustive list of the tooth-marked megamammal bone remains present in the various historical collections (19-20th centuries) in Europe and South America, is to make a very good state of the art on the various carnivores which were involved in the Pampas ecosystem during Pleistocene (species, ecology, ethology and type of bone modifications). Thus, although based on scarce elements, this work deserves to be published. Nevertheless, I submit to the authors a few minor issues and recommendations that need to be addressed. Authors would be able to consider these points before to re-submit their manuscript.
We also provided our comments directly on the proofs (PDF).

Annotated reviews are not available for download in order to protect the identity of reviewers who chose to remain anonymous.

·

Basic reporting

The article needs some editing for English language, grammar, and phrasing, which I have done on the World document version.
The lighting on Figures 3, 6, and 7 could be much improved to allow the reader to see the bone damage better.

Experimental design

No Comments.

Validity of the findings

The authors measure the tooth marks but do not compare those measurements to any published experimental tooth mark measurements (e.g. http://www.sciencedirect.com/science/article/pii/S0305440309002726) - this seems like a serious omission.

I don't understand why the authors dismiss Smilodon and Puma as possible bone modifiers - please see http://www.sciencedirect.com/science/article/pii/S0047248485710743 for Homotherium (relative of Smilodon) and http://www.sciencedirect.com/science/article/pii/S1040618215302536 for Puma.

Additional comments

It would be very useful to note the animal size (using some sort of standard classification system - I am familiar with the one proposed by Bunn 1992 for African animals). The size of the prey is quite relevant to the kind of damage the carnivore can inflict.

A table with a list of all the fossils with damage and descriptions of that damage would be very useful to the reader.

---

## Round 0.2 · accepted · Accept

While in production, please, try and remove the black background in Figs 4 and 6.

Reviewer 2 ·

Basic reporting

English editing done.
Figures have been added and modified as requested.

Experimental design

Methods and Material parts have been reorganized.
Data and statistical analyses about tooth-marks dimensions have been added to this study.

Validity of the findings

Some concluding remarks have been tempered as previously asked.

·

Basic reporting

The revised manuscript has much better English and is better organized and structured.

Experimental design

Good - no further comment.

Validity of the findings

Good - no further comment.

Additional comments

I commend the authors on the substantial amount of work they clearly invested to significantly improve this manuscript. I am comfortable with it being published in its present form, although it would be ideal if all photographs could have a white background (currently Figures 4 and 6 still have black backgrounds).